# A Comparative Study of Meta-Modeling for Response Estimation of Stochastic Nonlinear MDOF Systems Using MIMO-NARX Models

**Menghui Chen** [1], **Xiaoshu Gao** [2], **Cheng Chen** [3], **Tong Guo** [1,*] **and Weijie Xu** [1]

1   Key Laboratory of Concrete and Prestressed Concrete Structures, Ministry of Education, Southeast University, Nanjing 210096, China
2   School of Civil Engineering, Shandong University, Jinan 250061, China
3   School of Engineering, San Francisco State University, San Francisco, CA 94132, USA
*   Correspondence: guotong@seu.edu.cn

**Abstract:** Complex dynamic behavior of nonlinear structures makes it challenging for uncertainty analysis through Monte Carlo simulations (MCS). Surrogate modeling presents an efficient and accurate computational alternative for a large number of MCS. The previous study has demonstrated that the multi-input multi-output nonlinear autoregressive with exogenous input (MIMO-NARX) model provides good discrete-time representations of deterministic nonlinear multi-degree-of-freedom (MDOF) structural dynamic systems. Model order reduction (MOR) is executed to eliminate insignificant modes to reduce the computational burden due to too many degrees of freedom. In this study, the MIMO-NARX strategy is integrated with different meta-modeling techniques for uncertainty analysis. Different meta-models including Kriging, polynomial chaos expansion (PCE), and arbitrary polynomial chaos (APC) are used to surrogate the NARX coefficients for system uncertainties. A nine-DOF structure is used as an MDOF dynamic system to evaluate different meta-models for the MIMO-NARX. Good fitness of statistical responses is observed between the MCS results of the original system and all surrogated MIMO-NARX predictions. It is demonstrated that the APC-NARX model with the advantage of being data-driven is the most efficient and accurate tool for uncertainty quantification of nonlinear structural dynamics.

**Keywords:** nonlinear autoregressive with exogenous input (NARX); multi-degree-of-freedom; nonlinear structure; uncertainty analysis; meta-model

## 1. Introduction

Uncertainty analysis in structural engineering has attracted considerable research efforts in recent years [1–4]. It accounts for the influence of various uncertain factors including material property, external load, ground motion excitation, and experimental measurement, therefore is important for the design and optimization of structures and experiments to meet the performance target of robustness or reliability and to reduce the adverse impact of uncertainty propagation on system performance. Monte Carlo simulation (MCS), as the most direct and effective method for uncertainty analysis [5,6], however, requires significant computational efforts to obtain the statistical information of the uncertain outputs. Meta modeling also known as surrogate, has recently emerged as an efficient method for uncertainty analysis. The meta-models are trained by a number of samples of the original model to approximately predict the response quantities of interests in the entire sample space of uncertain inputs. The most commonly used meta-models include Kriging [7], polynomial chaos expansion (PCE) [8], arbitrary polynomial chaos expansion (APC) [9], etc.

Engineering structures generally involve nonlinear complex mechanical behavior related to the time history-dependent states of responses. Required computational time

significantly increases for accurate dynamic time-history simulations. The complex behavior exhibited by nonlinear systems brings great challenges to uncertainty analysis due to the requirement of a large number of samples for MCS. Dynamic analysis of a deterministic nonlinear structure can be considered as solving differential equations in the discrete-time domain. Nonlinear autoregressive with exogenous input (NARX) models provide good discrete-time representations of system dynamics as a function of previous response values as well as current and/or previous input signals [10,11]. The NARX model thus has been successfully applied for describing nonlinear dynamic behavior for single-degree-of-freedom (SDOF) systems [12]. In the case of multi-degree-of-freedom (MDOF) systems, the single-input single-output (SISO) strategy establishes a separate NARX model for each output of the MDOF system. Multi-input multi-output (MIMO) formulation considers the inevitable response coupling between the multiple outputs of the NARX model [13,14]. Nevertheless, a large number of degrees of freedom imposes significant computational demands for building the NARX model through the MIMO strategy. The model order reduction (MOR) method is usually utilized to reduce the degrees of freedom for the MIMO-NARX model, which has been demonstrated to significantly improve computational efficiency with little effect on its accuracy [15–17].

The NARX model can capture the nonlinear dynamic behavior of the deterministic system. A stochastic NARX model is characterized by a set of specified NARX model terms and associated random coefficients [18]. Once the NARX model is built from a series of nonlinear dynamic responses, the NARX coefficients need to be calibrated based on the specific system conditions including structural property and external excitation. This means that the NARX coefficients vary for different nonlinear dynamic behavior due to system uncertainties such as ground motions and structural properties. In uncertainty analysis, the nonlinear dynamic behavior of an MDOF system from a large number of MCS is difficult to obtain due to the unbearable computational cost of dynamic analysis. It means that the NARX coefficients cannot be calibrated directly by the target dynamic behaviors. Meta models have been used to govern the uncertainties in the nonlinear system represented by SISO-NARX in previous research [18,19]. Different from the Li et al. [14] method of adopting the constant mean value of the MIMO-NARX coefficients, this study builds different meta-models to surrogate the system uncertainties into the MIMO-NARX model coefficients for nonlinear system response prediction. More specifically, Kriging, PCE, and APC are utilized as meta-models to surrogate the NARX coefficients based on uncertain system parameters. The MIMO-NARX model is built in MOR-reduced coordinates using a series of numerical simulation samples to replicate the nonlinear dynamic behavior of the MDOF system with uncertainties. A nine-DOF structure is selected as a numerical case to evaluate the MIMO-NARX meta-models for nonlinear dynamic history prediction and to compare the effectiveness of different meta-models.

## 2. Brief Review of MIMO-NARX Modeling

### 2.1. MIMO-NARX Model for MDOF System

In structural dynamics, a nonlinear MDOF dynamic system can be expressed as:

$$\boldsymbol{M}\ddot{\boldsymbol{x}}(t) + \boldsymbol{C}\dot{\boldsymbol{x}}(t) + \boldsymbol{f}_r(\boldsymbol{x}(t)) = \boldsymbol{f}(t) \tag{1}$$

where $\boldsymbol{M}$ and $\boldsymbol{C}$ are the mass and damping matrix, respectively; $t$ denotes time; $\ddot{\boldsymbol{x}}(t)$, $\dot{\boldsymbol{x}}(t)$ and $\boldsymbol{x}(t)$ are response vectors of acceleration, velocity, and displacement, respectively; $\boldsymbol{f}_r$ is the vector of nonlinear hysteresis forces; $\boldsymbol{f}(t)$ is the external excitation force, which is often expressed in earthquake engineering as $\boldsymbol{f}(t) = -\boldsymbol{M}\boldsymbol{\iota}\ddot{x}_g(t)$ with an all-ones vector $\boldsymbol{\iota}$ and the seismic acceleration $\ddot{x}_g(t)$.

The nonlinear dynamic behavior of the MDOF system in Equation (1) can be captured in a NARX model by considering several steps of the current and past responses of input and output. A MIMO-NARX technique is adopted in this study to capture coupled nonlinear dynamics since a SISO-NARX model is unable to represent the response coupling between multiple outputs for general nonlinearity [13]. The current output $\boldsymbol{x}(t)$ of nonlin-

ear MDOF dynamic system depend on its past output values $[x(t - \Delta t), \ldots, x(t - n_o \Delta t)]$ with maximum time lag $n_o$ and time interval $\Delta t$, and current and past input values $[f(t), f(t - \Delta t), \ldots, f(t - n_i \Delta t)]$ with maximum time lag $n_i$ and can be expressed as:

$$x(t) = G(z(t)) + \epsilon(t) \tag{2}$$

where $z(t) = \left[ x^\mathrm{T}(t - \Delta t), \ldots, x^\mathrm{T}(t - n_o \Delta t), f^\mathrm{T}(t), \ldots, f^\mathrm{T}(t - n_i \Delta t) \right]$ is the regression vector of current and past values of input and output; $G(\cdot)$ represents the MIMO-NARX model to be identified based on the input and output; and $\epsilon(t)$ is vector-valued Gaussian error process [20]. This study adopts the linear-in-the-parameter form for the MIMO-NARX model which can be expressed as:

$$G(z(t)) = \Theta^\mathrm{T} g(z(t)) \tag{3}$$

where $g(\cdot) = \left[ g_1^\mathrm{T}(\cdot); g_2^\mathrm{T}(\cdot), \ldots, g_n^\mathrm{T}(\cdot) \right]^\mathrm{T}$ contains the NARX terms for the $n$ DOFs system; and $\Theta = \mathrm{diag}[\Theta_1, \ldots, \Theta_n]$ contains the NARX coefficients of the $n$ NARX models. For each degree of freedom, the output of a MIMO-NARX model can be formulated as:

$$x_j(t) = \Theta_j^\mathrm{T} g_j(z(t)) + \epsilon_j(t) \tag{4}$$

where $j$ represents the $j$th degree of freedom; $g_j(z(t_i))$ is the most relevant NARX terms selected from all potential NARX terms, that is, predetermined basis functions. While only the $j$th degree of freedom is considered in the SISO-NARX strategy, the MIMO-NARX strategy includes the input and output histories from all degrees of freedom.

Identification of the MIMO-NARX model requires the selection of NARX terms and calibration of their coefficient. An efficient approach [18] has been proposed for identifying the NARX model for SISO systems by implementing the least-angle regression (LARs) algorithm for structure determination and the ordinary least square (OLS) method for coefficient calibration. The LARs algorithm [21] is employed to select candidate NARX model terms by calculating the correlation between each potential model term and the system output [13]. The corresponding NARX coefficients can then be calculated based on the OLS method:

$$\Theta_{j,k} = \left[ Z_{j,k}^\mathrm{T} Z_{j,k} \right]^{-1} Z_{j,k}^\mathrm{T} X_j^\mathrm{T} \tag{5}$$

where $Z_{j,k}$ is the terms matrix of the $k$th candidate NARX model; and $X_j$ is the response time series of the $j$th degree of freedom. The most appropriate MIMO-NARX model is selected after the terms and coefficients of the candidate NARX models are determined. An appropriate error measure is conducted by decoupling the identification of the NARX models for each degree of freedom. For the $j$th MIMO-NARX model, only the outputs of the $j$th degree of freedom are estimated by recursively running the model, while the outputs of the other degrees of freedom are directly obtained from the simulation results. The error measure is defined as:

$$\widetilde{e}_{\mathrm{SE},j,k} = \frac{||X_j - \widetilde{X}_{j,k}||^2}{||X_j^T - \iota \mathbb{E}_t \left[ X_j \right] ||^2} \tag{6}$$

where $\mathbb{E}_t \left[ X_j \right]$ is the mean value of $X_j$; and $\widetilde{X}_{j,k}$ represents the response time series of the $j$th degree of freedom estimated from the $k$th candidate NARX model. The optimal model with model terms $g_j(\cdot)$ and coefficients $\Theta_j$ is selected as the candidate, which reaches a sufficiently small total error by satisfying a predefined threshold value. It is worth noting that as few as possible numbers of NARX terms have been found to avoid deleterious effects on the accuracy of the model, for example, overfitting and spurious dynamics [18,22,23].

### 2.2. Model Order Reduction

Although MIMO-NARX is suitable for multiple outputs, it is computationally expensive to establish the NARX model for each degree of freedom. The fact that the first few modes dominate the dynamic response of a multi-degree-of-freedom structure can be utilized for eliminating the burden due to a large number of degrees of freedom. The MOR method [15–17] thus can be utilized to lower the degrees of freedom of the MDOF nonlinear dynamic system in this study. Using a small number of lower structural modes corresponding to the number of DOF, an accurate approximation of the response history can be acquired. The response vector in the reduced space can be obtained through the coordinate transformation matrix and the response vector in the original space:

$$q(t) = \mathbf{\Phi}_{n_r}^{\mathrm{T}} x(t) \tag{7}$$

where $q(t)$ is the response in the reduced $n_r$-dimensional space; $x(t)$ is the response vector in the original $n$-dimensional space; $\mathbf{\Phi}_{n_r}^{\mathrm{T}}$ is the $n_r \times n$ coordinate transformation matrix with $n_r \ll n$;. The transformation matrix can be obtained by singular value decomposition (SVD) [24,25]. For the $n_s$ samples of the simulation results for the MDOF dynamic system under uncertainties, a snapshot matrix can be formulated as follows:

$$X = [x_1(t_1), \dots, x_1(t_{n_t}), \dots, x_{n_s}(t_1), \dots, x_{n_s}(t_{n_t})] \tag{8}$$

where $X \in \mathbb{R}^{n \times (n_t n_s)}$ contains $n_t$ time step responses of the $n$-DOF dynamic system. The SVD can then be used to decompose the snapshot matrix $X$ as:

$$X = U \mathbf{\Lambda} V^{\mathrm{T}} \tag{9}$$

where $U$ is a $n \times n$ orthonormal matrix that contains the left singular vectors of $X$; $V$ is the $(n_t n_s) \times (n_t n_s)$ orthonormal matrix that contains the right singular vectors; $\mathbf{\Lambda}$ is a $n \times (n_t n_s)$ pseudo-diagonal matrix with the singular values $\mathbf{\Lambda}(j,j) = \lambda_j$, $j = 1, \dots, n$. A computationally effective economy-size SVD is used in view of the extremely large size of $V$ with $(n_t n_s) \gg n$ [14]. The left singular vectors $U$ are the proper orthogonal decomposition (POD) modes.

Then the coordinate transformation matrix is defined as the first $n_r$ POD modes as:

$$\mathbf{\Phi}_{n_r} = [U_1, U_2, \dots, U_{n_r}] \tag{10}$$

where $[U_1, U_2, \dots, U_{n_r}]$ is the first $n_r$ columns of $U$; $n_r$ can be determined by the energy criterion which ensures the energy captured in the truncated representation of $x$ reaches $\eta$ of the total energy [17]:

$$\sum_{j=1}^{n_r} \lambda_j \geq \eta \sum_{j=1}^{n} \lambda_j \tag{11}$$

where $\eta$ is typically assumed to be close to 1, for example, 0.99. A properly selected $\eta$ can bring significant dimensional reduction to the system, that is, $n_r \ll n$. The transformation error of MOR can be measured by the comparison between the original response $x(t)$ from Equation (7) and the reverse-transformation response $x_-(t)$, which can be calculated as:

$$x_-(t) = \mathbf{\Phi}_{n_r} q(t) \tag{12}$$

### 2.3. NARX Model Pruning

The optimal NARX terms might still contain spurious terms due to the LARs approach selecting candidate terms through correlation analysis, which is not always a reflection of the contribution of a term to the model [22]. An iterative error-based pruning procedure [14] is adopted in this study to progressively delete the most deleterious NARX term at each iteration until a user-defined error threshold is reached. Each iteration intends to remove one unique term from the current MIMO-NARX model, and the coefficients of each trial model are recalculated through the OLS method. To evaluate the performance of trial

models, an error indicator is defined to describe the accuracy of the MIMO-NARX model based on the responses of the $m$th DOF in the original coordinate:

$$\hat{e}_{\text{SE},M} = \frac{||X_m - \Phi_{n_r}^m \hat{Q}||^2}{||X_m^T - \iota \mathbb{E}_t [X_m]||^2} \tag{13}$$

where $m$ is the DOF of interest; $X_m$ is the response time series of the $m$th DOF; $\Phi_{n_r}^m$ is the $m$th row of the transformation matrix $\Phi_{n_r}$; $\hat{Q}$ is the response matrix predicted by the MIMO-NARX model. The accuracy of the current MIMO-NARX model in each iteration can be evaluated by the mean value of error indicators overall training samples $\bar{e}_{\text{SE},m}$. While the error indicator for each trial model can be denoted as $\bar{e}'_{\text{SE},m}$. The current MIMO-NARX model is replaced by the optimal trial model with $\min\left\{\bar{e}'_{\text{SE},m} - \bar{e}_{\text{SE},m}\right\}$ in each iteration. The next iteration of the pruning process then proceeds with the new MIMO-NARX model serving as the current model until the predefined error threshold is satisfied, that is, $\bar{e}'_{\text{SE},m} - \bar{e}_{\text{SE},m} > \hat{E}$.

## 3. Meta-Modeling Techniques for Surrogating NARX Model Coefficients

In this study, meta-models are used to surrogate the NARX model coefficients to account for system uncertainties, which can be formulated as the $d$-dimensional input random variables $\boldsymbol{\xi} = [\xi_1, \xi_2, \ldots, \xi_d]^T \in \mathbb{R}^d$. System outputs can be generalized as $\boldsymbol{Y} = [y(\boldsymbol{\xi}_1), y(\boldsymbol{\xi}_2), \ldots, y(\boldsymbol{\xi}_N)]^T$ for $N$ groups of training samples. Three different meta-models are evaluated and compared in this study including Kriging, PCE, and APC.

### 3.1. Kriging Meta-Model

Kriging uses a weighted linear combination of all observed output values to estimate the system output of a random function or random process as a Gaussian process. These weights are generally described by the specified correlation function and are based on the distances between the location to be predicted and the locations already observed. The Kriging meta-model can be mathematically described as [26]:

$$y(\boldsymbol{\xi}) = \boldsymbol{\beta}^T f(\boldsymbol{\xi}) + \varepsilon(\boldsymbol{\xi}), \ \boldsymbol{\beta}^T f(\boldsymbol{\xi}) = \sum_{i=1}^{D} \beta_i f_i(\boldsymbol{\xi}) \tag{14}$$

where $\boldsymbol{\beta}^T f(\boldsymbol{\xi})$ is the mean value of the Gaussian process; $f_i(\boldsymbol{\xi})$ is the $i$th polynomial basis function with corresponding coefficients $\beta_i$; $D$ represents the total number of basis function terms which depends on the degree of the polynomials. The residual term $\varepsilon(\boldsymbol{\xi})$ is assumed to have zero mean and the following covariance:

$$\text{Cov}\left[\varepsilon(\boldsymbol{\xi}_i), \ \varepsilon(\boldsymbol{\xi}_j)\right] = \sigma_Z^2 \mathcal{R}\left(\theta, \ \boldsymbol{\xi}_i, \ \boldsymbol{\xi}_j\right) \tag{15}$$

where $\boldsymbol{\xi}_i$ and $\boldsymbol{\xi}_j$ represent the two different sample inputs; $\sigma_Z^2$ is the process variance; and $\mathcal{R}\left(\theta, \ \boldsymbol{\xi}_i, \ \boldsymbol{\xi}_j\right)$ represents the correlation function with parameter $\theta$. Several auto-correlation functions have been used in the literature [26–29]. In this study, the Gaussian correlation function is adopted, which can be expressed as:

$$\mathcal{R}\left(\boldsymbol{\xi}_i, \boldsymbol{\xi}_j\right) = \prod_{k=1}^{d} \exp[-\theta_k(\xi_{i,k} - \xi_{j,k})^2] \tag{16}$$

where the parameters $\theta_k$ are hyper-parameters of auto-correlation functions.

The Kriging model predicts the response for untried input $\boldsymbol{\xi}_u$ through the best linear unbiased predictor:

$$\hat{y}(\boldsymbol{\xi}_u) = \hat{\boldsymbol{\beta}}^T f(\boldsymbol{\xi}_u) + \mathfrak{R}^T \mathcal{R}^{-1}\left(\hat{\boldsymbol{\beta}}^T f(\boldsymbol{\xi}_u)\right) \tag{17}$$

$$\sigma_{\hat{M}}^2(\boldsymbol{\xi}_u) = \hat{\sigma}_Z^2 \left(1 - \begin{bmatrix} f^T(\boldsymbol{\xi}_u) & \mathfrak{R}^T(\boldsymbol{\xi}_u) \end{bmatrix} \begin{bmatrix} \mathbf{0} & \mathcal{L} \\ \mathcal{L} & \mathcal{R} \end{bmatrix}^{-1} \begin{bmatrix} f(\boldsymbol{\xi}_u) \\ \mathfrak{R}(\boldsymbol{\xi}_u) \end{bmatrix}\right) \tag{18}$$

where $\mathfrak{R}(\boldsymbol{\xi}_u) = [\mathcal{R}(\boldsymbol{\xi}_u, \boldsymbol{\xi}_1), \mathcal{R}(\boldsymbol{\xi}_u, \boldsymbol{\xi}_2), \ldots, \mathcal{R}(\boldsymbol{\xi}_u, \boldsymbol{\xi}_N)]^T$ defines the correlation matrix between untried sample point $\boldsymbol{\xi}_u$ and training samples; $\mathcal{L}$ is the matrix of basis functions at training points of dimension $N \times D$; and $\mathcal{R}$ is the correlation matrix for training samples.

The model parameters $\hat{\boldsymbol{\beta}}$ and $\hat{\sigma}_Z^2$, and the hyper-parameters $\theta$ can be determined using the maximum-likelihood estimation (MLE) [27,28]:

$$\hat{\boldsymbol{\beta}} = \left(\mathcal{L}^T \mathcal{R}^{-1} \mathcal{L}\right)^{-1} \mathcal{L}^T \mathcal{R}^{-1} \boldsymbol{Y} \tag{19}$$

$$\hat{\sigma}_Z^2 = \frac{1}{N} \left(\boldsymbol{Y} - \mathcal{L}\hat{\boldsymbol{\beta}}\right)^T \mathcal{R}^{-1} \left(\boldsymbol{Y} - \mathcal{L}\hat{\boldsymbol{\beta}}\right) \tag{20}$$

More details of the Kriging meta-model can be referred to in the work of Santner et al. [28]. The MATLAB-based toolbox DACE [30] is used in this study for the Kriging model.

### 3.2. PCE Meta-Model

PCE is a weighted linear combination of orthogonal polynomial basis functions of input random variables [31] and has been widely used in many fields due to its solid mathematical foundation and good performance. For the $d$-dimensional input random variables $\boldsymbol{\xi} = [\xi_1, \xi_2, \ldots, \xi_d]^T \in \mathbb{R}^d$, the single system output $y$ can be expanded through polynomial chaos expansion as:

$$y(\boldsymbol{\xi}) = \sum_{i=0}^{D-1} C_i \Phi^{(i)}(\boldsymbol{\xi}) + \varepsilon \tag{21}$$

where $\Phi^{(i)}(\boldsymbol{\xi})$ is the $i$th orthogonal multivariate basis function composed of Winer–Askey polynomials; the terms number $D = (d+r)!/(d!r!)$ with the PCE order $r$; and $\varepsilon$ is the truncation error. The coefficients $\boldsymbol{c} = [c_0, c_1, \ldots, c_{D-1}]^T$ can be determined through least square regression as:

$$\boldsymbol{c} = \left(\boldsymbol{A}^{\mathrm{T}} \boldsymbol{A}\right)^{-1} \boldsymbol{A}^{\mathrm{T}} \boldsymbol{Y} \tag{22}$$

$$\boldsymbol{A} = \begin{bmatrix} \Phi^{(0)}(\boldsymbol{\xi}_1) & \Phi^{(1)}(\boldsymbol{\xi}_1) & \cdots & \Phi^{(D-1)}(\boldsymbol{\xi}_1) \\ \Phi^{(0)}(\boldsymbol{\xi}_2) & \Phi^{(1)}(\boldsymbol{\xi}_2) & \cdots & \Phi^{(D-1)}(\boldsymbol{\xi}_2) \\ \vdots & \vdots & \ddots & \vdots \\ \Phi^{(0)}(\boldsymbol{\xi}_N) & \Phi^{(1)}(\boldsymbol{\xi}_N) & \cdots & \Phi^{(D-1)}(\boldsymbol{\xi}_N) \end{bmatrix}, \boldsymbol{Y} = \begin{bmatrix} Y(\boldsymbol{\xi}_1) \\ Y(\boldsymbol{\xi}_2) \\ \vdots \\ Y(\boldsymbol{\xi}_N) \end{bmatrix} \tag{23}$$

where $\boldsymbol{A}$ is the matrix of orthonormal multivariate polynomials; $\boldsymbol{Y}$ is the real value vector of the sample output; and $N$ is the number of samples, which is suggested to be no less than twice the number of expansion terms $D$. In this study, the MATLAB-based UQLab toolbox [32] is utilized to construct the PCE model for the NARX model coefficients obtained from the least square method. The orthogonal polynomials are chosen based on the distribution of input variables. When the system has multiple outputs under uncertain inputs, one PCE surrogate model needs to be constructed for each system output.

### 3.3. APC Meta-Model

APC uses a data-driven technique to build the optimal orthogonal polynomial basis function without assuming the distribution types of inputs, which extends PCE to apply to arbitrary distributions of random input variables [9]. The system output $y$ can be expressed as:

$$y(\boldsymbol{\xi}) = \sum_{i=0}^{D-1} c_i P^{(i)}(\boldsymbol{\xi}) + \varepsilon \tag{24}$$

where $D$ is the number of APC expansion terms; $P^{(i)}(\boldsymbol{\xi})$ is the optimal orthogonal multivariate polynomial obtained through data-driven technique; $c_i$ is the corresponding expansion

coefficient obtained from the least square method; The *i*th term orthogonal multivariate polynomial $P^{(i)}(\boldsymbol{\xi})$ is multiplied by univariate orthogonal polynomials as:

$$P^{(i)}(\boldsymbol{\xi}) = \prod_{j=1}^{d} P_j^{(l_{i,j})}(\xi_j) \tag{25}$$

$$\sum_{j=1}^{d} l_{i,j} = 0, 1, \ldots, r \tag{26}$$

where $P_j^{(l_{i,j})}(\xi_j)$ represents the $l_{i,j}$th order univariate orthogonal polynomial of the *j*th dimensional input variable; the sum of the orders $l_{i,j}$ of the *d* univariate orthogonal polynomials ranges from 0 to *r*, and *r* represents the highest order of the multivariate APC models. The univariate orthogonal polynomial $P_j^{(l)}(\xi_j)$ is obtained based on statistical moments as in the following expression:

$$P_j^{(l)}(\xi_j) = \sum_{k=0}^{l} p_k^{(l)} X^k \tag{27}$$

$$\begin{bmatrix} \mu_0 & \mu_1 & \cdots & \mu_l \\ \mu_1 & \mu_2 & \cdots & \mu_{l+1} \\ \vdots & \vdots & \ddots & \vdots \\ \mu_{l-1} & \mu_l & \cdots & \mu_{2l-1} \\ 0 & 0 & \cdots & 1 \end{bmatrix} \begin{bmatrix} p_0^{(l)} \\ p_1^{(l)} \\ \vdots \\ p_{l-1}^{(l)} \\ p_l^{(l)} \end{bmatrix} = \begin{bmatrix} 0 \\ 0 \\ \vdots \\ 0 \\ 1 \end{bmatrix} \tag{28}$$

where the interior coefficients $p_k^{(l)}$ of the univariate orthogonal polynomial are calculated from $\mu_0$ to $\mu_{2l-1}$, that is, the statistical moment with order 0 to $(2l - 1)$ of the sample data of the univariate input variable $\xi_j$. APC meta-model is directly obtained from the sample data and has demonstrated faster error convergence with the increasing order and better accuracy than PCE of the same order [33].

## 4. Training of MIMO-NARX Meta-Models

The MIMO-NARX model is trained offline using numerical simulation data of the structural system in this study. The procedure includes generating sample data from the nonlinear dynamic history simulations, training MIMO-NARX terms using reduced coordinate from model order reduction, and surrogating NARX coefficients through meta-models using the uncertain parameters of the nonlinear dynamic system.

### 4.1. Nine-DOF Shear Structure

A nine-DOF shear structure is used as the MDOF nonlinear dynamic system in this study, which is simplified from a nine-story benchmark frame [34] as shown in Figure 1a. The first three natural frequencies and mode shapes of this nine-DOF structure are shown in Figure 1b.

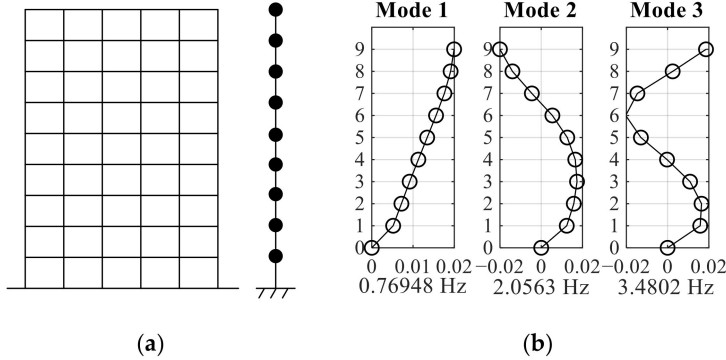

(a)                        (b)

**Figure 1.** Nine-DOF shear structure simplified from nine-story frame. (**a**) Target structure; (**b**) Natural frequencies and modes.

The equation of motion of the nine-DOF under seismic excitation can be expressed by Equation (1) and the nonlinear restoring vector $f_r$ is emulated using the Bouc–Wen model [35,36], which can be formulated as:

$$f_r = k_{bw}[\alpha_{bw}x + (1 - \alpha_{bw})z] \tag{29}$$

$$\dot{z} = A_{bw}\dot{x} - \gamma_{bw}\dot{x}|z|^{n_{bw}} - \beta_{bw}|\dot{x}||z|^{n_{bw}-1}z \tag{30}$$

where $f_r$ is the hysteresis force for each story; $x$ is the inter-story displacement between two neighboring floors; and the Bouc–Wen model parameters ($\alpha_{bw}$, $A_{bw}$, $\gamma_{bw}$, $\beta_{bw}$, and $n_{bw}$) are listed in Table 1. Under a typically selected ground motion excitation, moderate nonlinearity can be observed in Figure 2 for the Bouc–Wen models of each story.

**Table 1.** Bouc–Wen model parameters.

| Floor | $k_{bw}$ (N/mm) | $\alpha_{bw}$ | $A_{bw}$ | $\gamma_{bw}$ | $\beta_{bw}$ | $n_{bw}$ |
|-------|-----------------|---------------|----------|---------------|--------------|----------|
| 1 | $2.9793 \times 10^5$ | 0.01 | 1 | 0.02 | 0.02 | 0.01 |
| 2 | $7.3787 \times 10^5$ | 0.01 | 1 | 0.1 | 0.1 | 0.01 |
| 3 | $6.9252 \times 10^5$ | 0.01 | 1 | 0.1 | 0.1 | 0.01 |
| 4 | $6.0177 \times 10^5$ | 0.01 | 1 | 0.1 | 0.1 | 0.01 |
| 5 | $5.2397 \times 10^5$ | 0.01 | 1 | 0.08 | 0.08 | 0.01 |
| 6 | $4.4108 \times 10^5$ | 0.01 | 1 | 0.06 | 0.06 | 0.01 |
| 7 | $3.6986 \times 10^5$ | 0.01 | 1 | 0.06 | 0.06 | 0.01 |
| 8 | $3.4706 \times 10^5$ | 0.01 | 1 | 0.08 | 0.08 | 0.01 |
| 9 | $3.2748 \times 10^5$ | 0.01 | 1 | 0.1 | 0.1 | 0.01 |

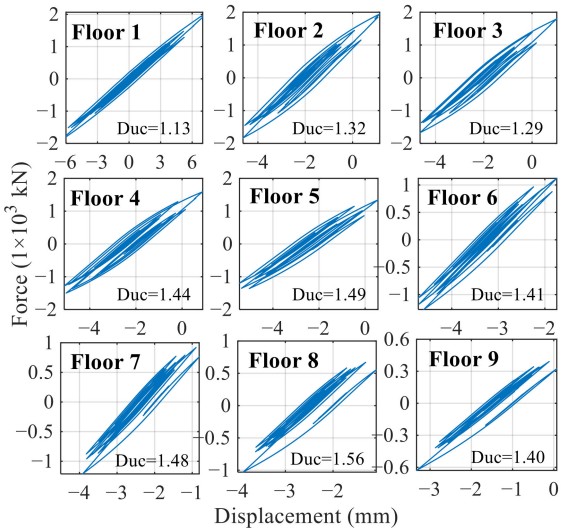

**Figure 2.** Hysteresis behavior of Bouc–Wen models (a typical sample).

### 4.2. Uncertainties in Ground Motion and Structural Properties

The dynamic responses are affected by the system uncertainties in ground motion and structural properties. By varying ground motions and structural parameters, structural responses can be obtained and used as training samples. The stochastic ground motion model [37] is adopted in this study to generate ground motion uncertainties. Each ground motion is determined by six parameters $\left(\bar{I}_a, D_{5-95}, t_{mid}, \omega_{mid}, \omega', \zeta_f\right)$ and white noise. In this study, $\omega_{mid}$ and $\zeta_f$, which control the filter frequency and damping ratio, respectively, are selected as uncertain parameters of ground motion while the other parameters ($\bar{I}_a$, $D_{5-95}$, $t_{mid}$, $\omega'$) are set as (0.0314 s·g, 11.23 s, 7.85 s, −0.04 Hz/s). It is worth noting that the white noise remains unchanged to generate seismic acceleration time histories. A typical acceleration time history is shown in Figure 3.

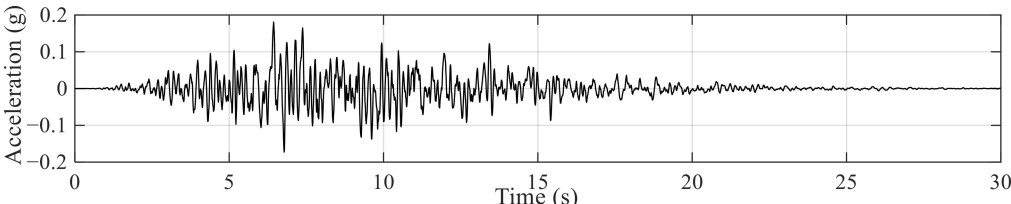

**Figure 3.** Time history of stochastic ground motion (a typical sample).

A total of $n_s = 200$ parameter sets of $(\omega_{mid}, \zeta_f)$ are generated following the distribution in Figure 4a,b, which leads to 200 stochastic ground motions for MIMO-NARX training. Figures 4c and 5 present the distribution of peak ground accelerations and the mean response spectra of these 200 acceleration time histories, respectively.

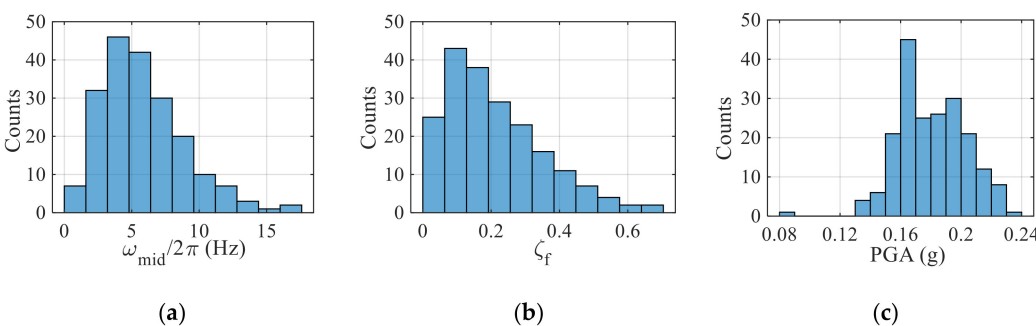

(**a**)             (**b**)             (**c**)

**Figure 4.** Uncertainties of ground motions (200 samples). (**a**) Filter frequency $\omega_{mid}$; (**b**) Filter damping ratio $\zeta_f$; (**c**) Peak ground acceleration.

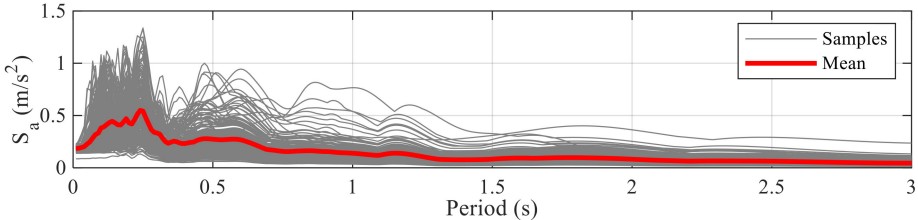

**Figure 5.** Response spectra of ground motions with damping ratio 0.05 (200 samples).

The concentrated masses of each story are considered structural uncertainties, which reflects the uncertainty of live load or temporary load changes in the building. The masses of floor 1, floor 2~8, and floor 9 are assumed to follow the Gaussian distribution as shown in Figure 6.

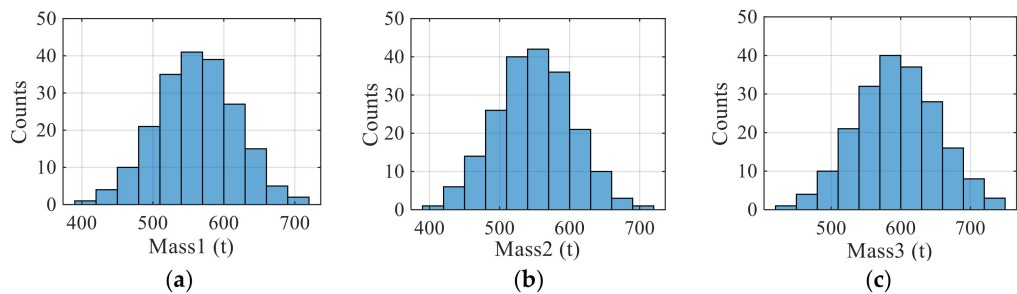

(**a**)             (**b**)             (**c**)

**Figure 6.** Variable mass for structural uncertainties (200 samples). (**a**) Floor 1; (**b**) Floor 2~8; (**c**) Floor 9.

The two ground motion parameters along with the three structural parameters are considered uncertain input variables in this study and are listed in Table 2. The uncertain variables of the dynamic system can be generated randomly from the distribution in Table 2

for dynamic time history analysis. The sample data used for training the MIMO-NARX model are extracted from the current and past state values of structural responses.

**Table 2.** Marginal distributions of uncertainties.

| Variable | Distribution Type | Bounds | Mean | Standard Deviation |
|---|---|---|---|---|
| $\omega_{mid}/2\pi$ | Gamma | $[0, +\infty]$ | 5.87 | 3.11 |
| $\zeta_f$ | Beta | $[0.02, 1]$ | 0.213 | 0.143 |
| Mass 1 | Gaussian | $[392.77, 729.43]$ | 561.1 | 56.11 |
| Mass 2 | Gaussian | $[384.58, 714.22]$ | 549.4 | 54.94 |
| Mass 3 | Gaussian | $[416.08, 772.72]$ | 594.4 | 59.44 |

*4.3. MIMO-NARX Model Training*

Firstly, the MOR strategy is applied to transform the nine-DOF to the reduced coordinates for the convenience of MIMO-NARX modeling. The snapshot matrix $\mathbf{X}$ is extracted from the structural responses of the two hundred dynamic history simulations. The transformation matrix $\mathbf{\Phi}_{n_r}$ is obtained through SVD as the first $n_r$ POD modes according to Equations (9) and (10). The coordinate is transformed from the original order $n = 9$ to the reduced order $n_r = 4$, which is determined by the energy criteria in Equation (11). The fitness indicator of the nine-DOF response in original coordinates and reverse-transformation response is shown in Figure 7a, in which the red line indicates the median, the blue box indicates the 25th and 75th percentiles, and the red plus symbol indicates outliers. It can be observed from Figure 7a that the maximum error of MOR occurs in the first DOF because the value of absolute response of the first floor is the smallest compared with the other floors. For a randomly selected sample, the comparison in Figure 7b shows a good fitness of 92% between original and reverse-transformation velocity responses $\dot{x}(t)$ and $\dot{x}\_(t)$ for the first floor. This implies that the error introduced by MOR is negligible in this study.

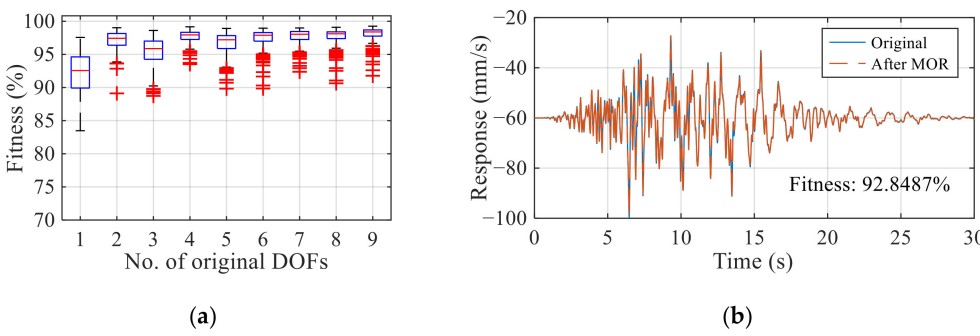

(a)                    (b)

**Figure 7.** MOR error between the original and reverse-transformation responses. (**a**) Fitness for nine DOFs (200 samples); (**b**) 1st DOF response (a random sample).

To capture the nonlinear dynamic behavior of the MDOF system, the MIMO-NARX model is trained from the $n_s = 200$ dynamic history simulation samples. Multiple inputs and outputs in the reduced coordinates are then used to build the MIMO-NARX model to avoid significant computational efforts associated with the original number of DOFs. The simulation samples are first examined by a predefined threshold value to screen responses of interest that exhibit suitable nonlinearity [18]. NARX candidates are identified by the LARs algorithm and the corresponding term coefficients are calculated by the OLS approach as shown in Figure 8. The optimal NARX model is selected based on the minimum mean errors of these candidates and terms pruning is then applied to improve precision. If the accuracy requirement is not satisfied for the optimal NARX model, some measures to improve NARX accuracy need to be taken such as redesigning the formulation of the NARX full model.

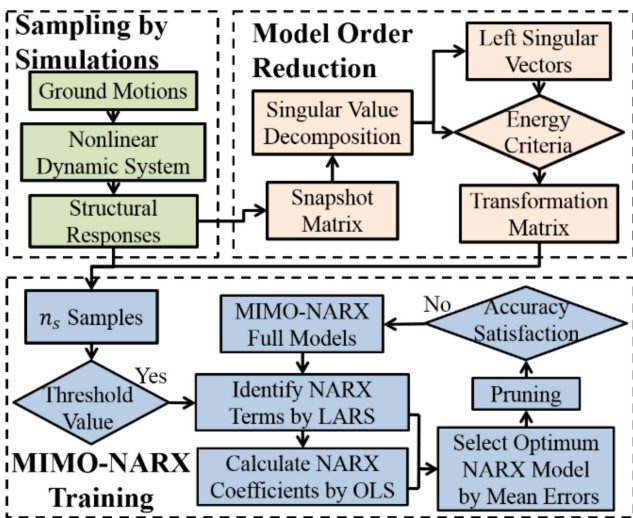

**Figure 8.** Procedure for MIMO-NARX training.

In this study, the MIMO-NARX full model is designed as shown in Figure 9. The ground motion accelerations are considered as the input, while responses of the $n_r = 4$ MOR reduced coordinates are used as the outputs. The maximum time lag of $n_i = n_o = 4$ is used for the input and output in the NARX model to capture the memories. The current value for each output is replicated by one MIMO-NARX model considering the coupling of the past signals of all four outputs.

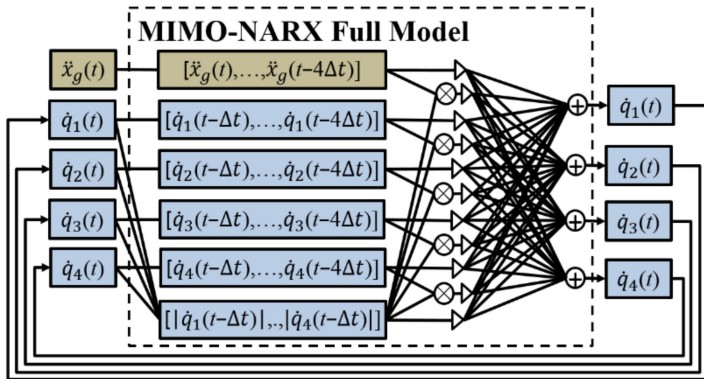

**Figure 9.** Structure of MIMO-NARX full model.

The velocity responses in the reduced coordinates are selected as NARX outputs because previous studies showed that it is more effective to describe nonlinear systems with respect to velocity. The full NARX model consists of potential NARX terms including seismic acceleration from 0 to $l$ steps forward $\ddot{x}_g(t - l\Delta t)$, velocity responses of the reduced coordinates from 1 to $l$ steps forward $\dot{q}_j(t - l\Delta t)$, one step ahead absolute value of velocity responses of the reduced coordinates $\left|\dot{q}_j(t - \Delta t)\right|$, $\left|\dot{q}_j(t - \Delta t)\right|\dot{q}_j(t - l\Delta t)$, $\left|\dot{q}_j(t - \Delta t)\right|\ddot{x}_g(t - l\Delta t)$ and 1, with $l = 1, 2, 3, 4$, $j = 1, 2, 3, 4$, which leads to $5 + 4 \times 4 + 4 + 4 \times 4 \times 4 + 4 \times 5 + 1 = 110$ terms. It should be noticed that the absolute value terms directly use velocity responses of the reduced coordinates rather than velocity responses in the original coordinates $\mathbf{\Phi}_{n_r}^m \dot{q}(t - \Delta t)$ used in previous studies [14]. This results in a smaller

number of terms to avoid redundant terms, which brings the benefits of faster computation and almost similar accuracy. The final MIMO-NARX model can be expressed as follows:

$$
\begin{aligned}
\dot{q}_1(t) = y_{1,1}\ddot{x}_g(t) \quad &+ y_{1,2}\ddot{x}_g(t-\Delta t) + y_{1,3}\ddot{x}_g(t-2\Delta t) + y_{1,4}\ddot{x}_g(t-3\Delta t) \\
&+ y_{1,5}\ddot{x}_g(t-4\Delta t) + \cdots + y_{1,55}\big|\dot{q}_2(t-\Delta t)\big|\ddot{x}_g(t-4\Delta t) \\
&+ y_{1,56}\big|\dot{q}_3(t-\Delta t)\big|\ddot{x}_g(t-4\Delta t) + y_{1,57}\big|\dot{q}_4(t-\Delta t)\big|\ddot{x}_g(t-4\Delta t)
\end{aligned}
\tag{31}
$$

$$
\begin{aligned}
\dot{q}_2(t) = y_{2,1}\ddot{x}_g(t) \quad &+ y_{2,2}\ddot{x}_g(t-\Delta t) + y_{2,3}\ddot{x}_g(t-2\Delta t) + y_{2,4}\ddot{x}_g(t-3\Delta t) \\
&+ y_{2,5}\ddot{x}_g(t-4\Delta t) + \cdots + y_{2,60}\big|\dot{q}_2(t-\Delta t)\big|\ddot{x}_g(t-4\Delta t) \\
&+ y_{2,61}\big|\dot{q}_3(t-\Delta t)\big|\ddot{x}_g(t-4\Delta t) + y_{2,62}\big|\dot{q}_4(t-\Delta t)\big|\ddot{x}_g(t-4\Delta t)
\end{aligned}
\tag{32}
$$

$$
\begin{aligned}
\dot{q}_3(t) = y_{3,1}\ddot{x}_g(t) \quad &+ y_{3,2}\ddot{x}_g(t-4\Delta t) + y_{3,3}\dot{q}_2(t-3\Delta t) + y_{3,4}\dot{q}_2(t-4\Delta t) \\
&+ y_{3,5}\dot{q}_3(t-\Delta t) + \cdots + y_{3,27}\big|\dot{q}_2(t-\Delta t)\big|\ddot{x}_g(t) \\
&+ y_{3,28}\big|\dot{q}_3(t-\Delta t)\big|\ddot{x}_g(t) + y_{3,29}\big|\dot{q}_4(t-\Delta t)\big|\ddot{x}_g(t)
\end{aligned}
\tag{33}
$$

$$
\begin{aligned}
\dot{q}_4(t) = y_{4,1}\ddot{x}_g(t) \quad &+ y_{4,2}\ddot{x}_g(t-\Delta t) + y_{4,3}\ddot{x}_g(t-2\Delta t) + y_{4,4}\ddot{x}_g(t-3\Delta t) \\
&+ y_{4,5}\ddot{x}_g(t-4\Delta t) + \cdots + y_{4,47}\big|\dot{q}_2(t-\Delta t)\big|\ddot{x}_g(t-4\Delta t) \\
&+ y_{4,48}\big|\dot{q}_3(t-\Delta t)\big|\ddot{x}_g(t-4\Delta t) + y_{4,49}\big|\dot{q}_4(t-\Delta t)\big|\ddot{x}_g(t-4\Delta t)
\end{aligned}
\tag{34}
$$

where $y_{i,j}$ represents the NARX coefficient for the $j$th NARX term of the $i$th output. It can be observed that there are 57, 62, 29, and 49 terms for the four outputs $\dot{q}_1(t)$, $\dot{q}_2(t)$, $\dot{q}_3(t)$, and $\dot{q}_4(t)$, respectively. Higher order of DOF does not necessarily lead to more terms for the MIMO-NARX model.

### 4.4. Surrogating NARX Coefficients by Meta-Models

The MIMO-NARX model trained by the simulation samples is aimed to determine the optimal NARX candidate terms from Figure 9. The corresponding NARX coefficients need to be calibrated based on the system responses in the specific dynamic simulation case. For the $n_s$ = 200 simulation samples, the five parameters of ground motion and structural masses in Table 2 are considered uncertain factors affecting the dynamic responses. Therefore, the coefficients of the MIMO-NARX model are taken as output variables varying with the five input variables of system uncertainties. The meta-models are then built to surrogate the NARX coefficients using the five uncertain parameters under the condition that the responses of a large number of MCS are unknown as shown in Figure 10. This study explores three meta-models including Kriging, PCE, and APC. The order of 3 is used for PCE and APC as the recommended number of samples is twice the number of polynomial terms.

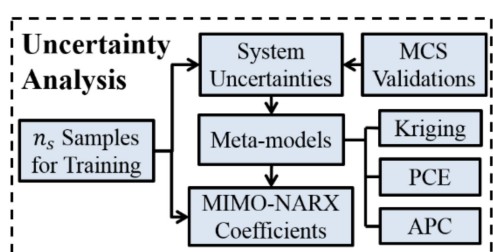

**Figure 10.** Meta-models to surrogate MIMO-NARX coefficients.

## 5. Validation of Surrogated MIMO-NARX Model

To explore the uncertainty propagation in a nonlinear dynamic system with MDOF, a total of 300 validation Monte Carlo simulations with different ground motions and structural properties are performed in this section by traditional dynamic history analysis using the CR algorithm [38]. The MIMO-NARX models obtained from the previous section are adopted to capture the nonlinear dynamic behavior while the corresponding coefficients are calibrated by the meta-models for each set of uncertainty variable values. Finally, structural responses are compared between the MIMO-NARX model prediction and MCS to evaluate its effectiveness.

An error evaluation criterion is introduced to measure the fitness between the predicted and original value and expressed as:

$$\text{Fitness} = \left(1 - \sqrt{\frac{\sum(x(i) - x^s(i))^2}{\sum(x(i) - \overline{x})^2}}\right) \times 100\% \qquad (35)$$

where $x(i)$ represent the original response from MCS with mean $\overline{x}$; and $x^s(i)$ represent the predicted response by the surrogated MIMO-NARX model.

### 5.1. Accuracy Assessment of MIMO-NARX Outputs in Reduced Coordinates

The MIMO-NARX model replicates four outputs of the order-reduced model in reduced coordinates as shown in Figure 11 for a typically selected ground motion with the four reduced coordinates presented from low to high. It can be observed in Figure 11 that the outputs of the MIMO-NARX model with the coefficients calibrated by Kriging, PCE, and APC meta-models are basically consistent with the MCS responses transformed to reduced coordinates. The matching accuracy of the four reduced coordinates, however, decreases with the increase of frequency due to the difficulty in capturing high-frequency nonlinearity as shown in Figure 12. For this specific selected ground motion, the APC meta-model has better accuracy for the fourth output prediction than that of Kriging and APC meta-models, while the three meta-models have nearly similar accuracy for the first three outputs.

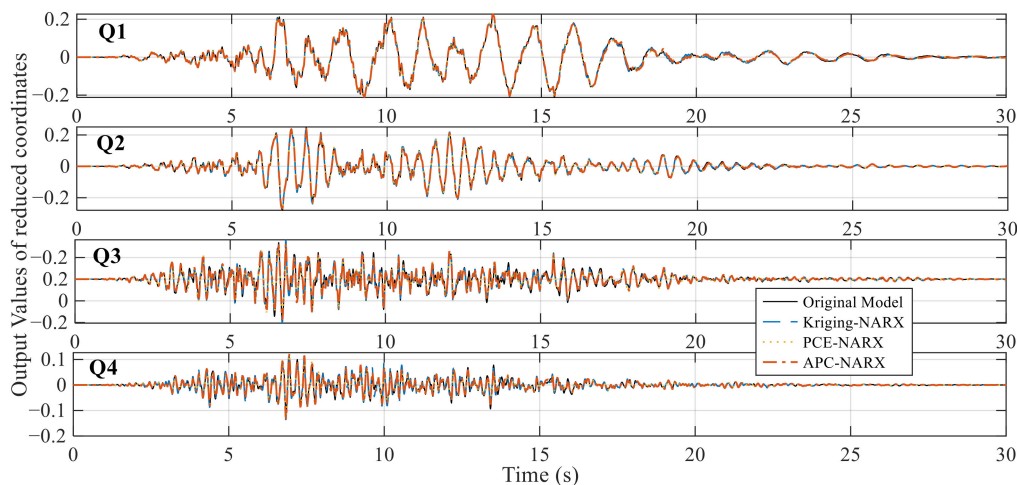

**Figure 11.** Time history of the four outputs in reduced coordinates (a typical sample).

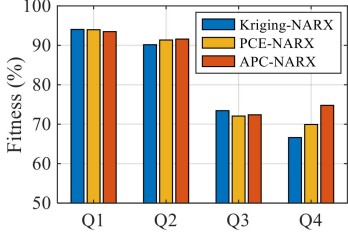

**Figure 12.** Fitness for the four outputs (a typical sample).

### 5.2. Accuracy Assessment of Absolute Structural Responses

Since the absolute velocity responses are used in NARX training to better capture nonlinearity, Figure 13a first presents the comparison of absolute velocity responses for the nine-DOF structure, which are obtained from the four outputs of the reduced coordinates from the MIMO-NARX model. Good fitness can be observed between Kriging/PCE/APC-NARX model prediction and MCS results. The absolute displacement responses are then

acquired through the integration of corresponding velocity responses and presented in Figure 13b. Slight errors can be observed at the end of time history analysis which can be attributed to integral error and error accumulation. Figure 14 compares the fitness index of absolute responses between three meta-models. Among the three meta-models, almost exactly the same fitness results can be observed for absolute velocity responses. Figure 14a shows that the first floor still has the lowest accuracy compared with the other floors, which is consistent with MOR errors in Figure 7. It can also be observed from Figure 14b that the Kriging meta-model has worse accuracy in absolute displacement prediction than the PCE and APC meta-models due to the drift errors at the end of the time history response. The drift errors are obvious in the lower and upper floors but less significant in the middle floors for the Kriging-NARX results of the selected ground motion.

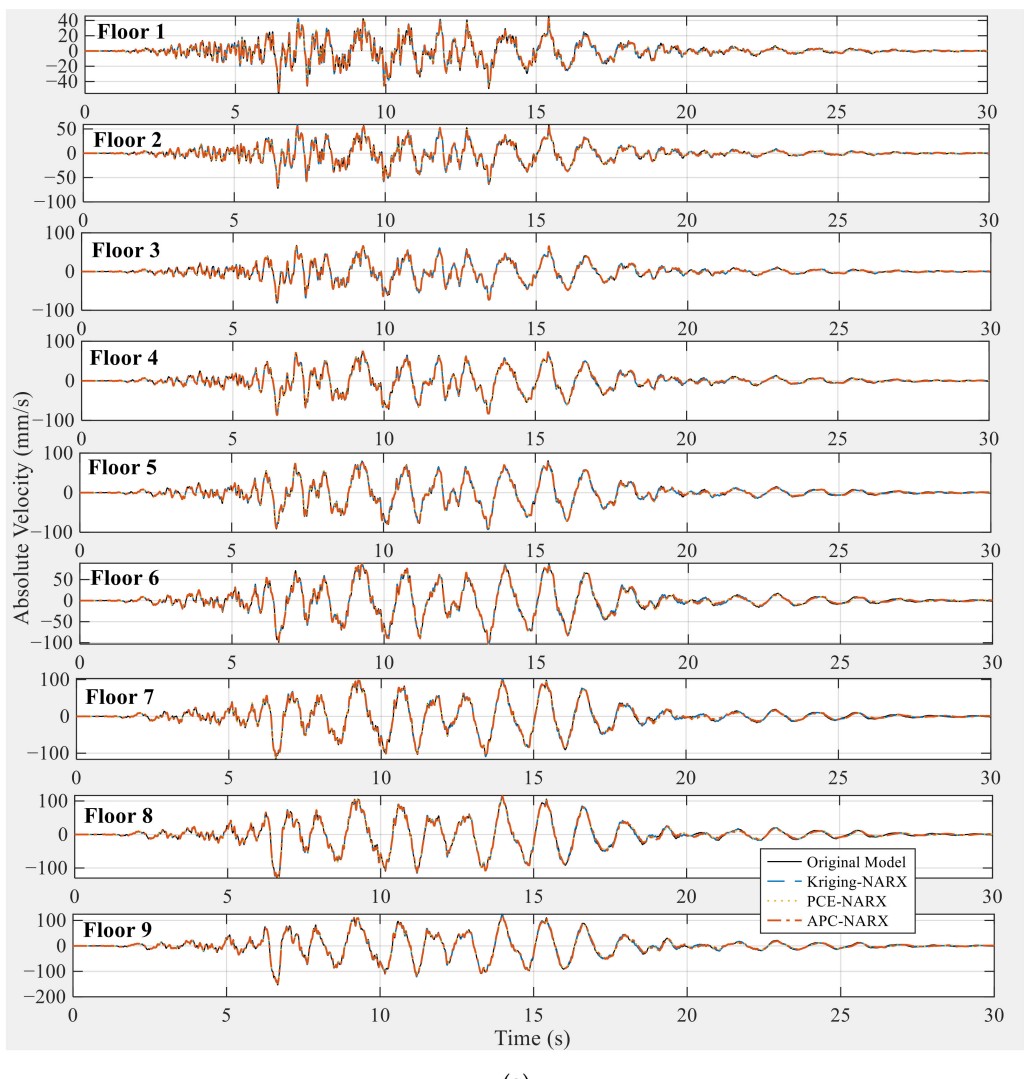

(**a**)

**Figure 13.** *Cont.*

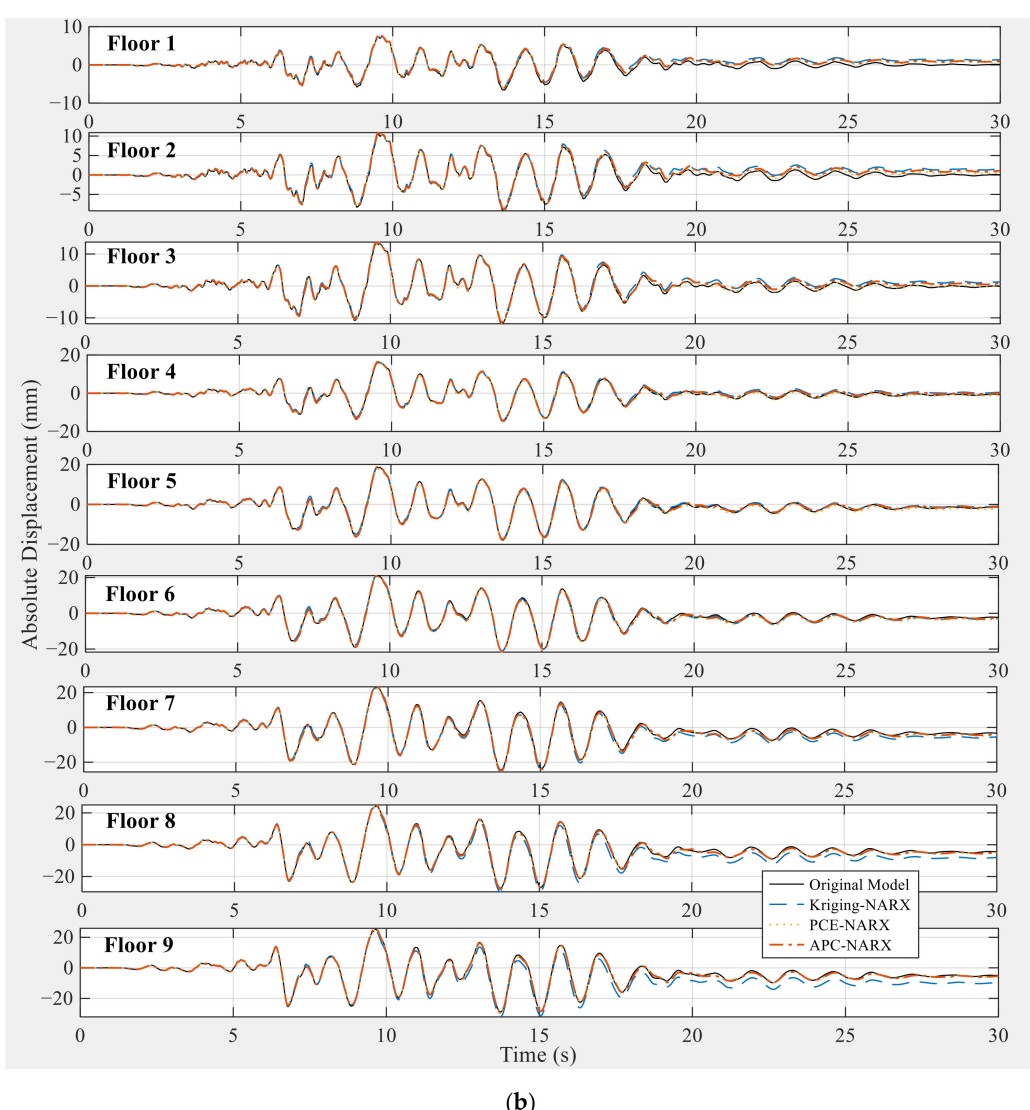

(**b**)

**Figure 13.** Time history of the absolute responses (a typical sample). (**a**) Velocity responses; (**b**) Displacement responses.

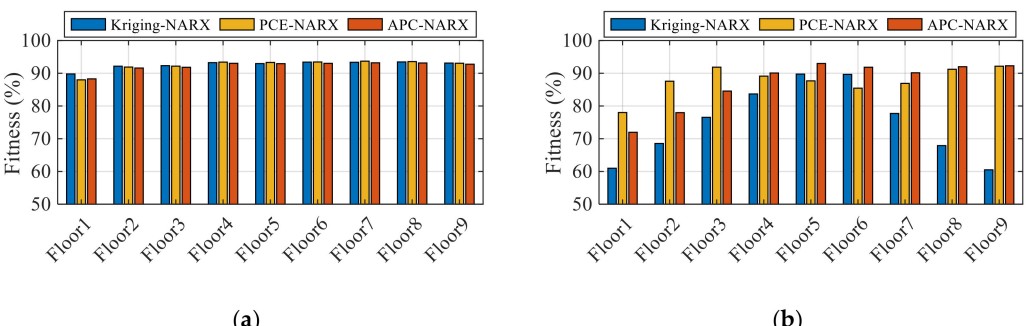

**Figure 14.** Fitness for the absolute responses (a typical sample). (**a**) Velocity responses; (**b**) Displacement responses.

### 5.3. Accuracy Assessment of Inter-Story Structural Responses

Inter-story velocity and displacement responses are further calculated from the absolute velocity and displacement responses of the nine-DOF structure. Figure 15a,b presents the comparison of inter-story velocity and displacement responses for the nine-DOF struc-

ture, respectively. As can be seen from the figures, the inter-story velocity curves still have good fitting accuracy, while the inter-story displacement curves show more obvious drift error from integral accumulation error, especially in the upper floors. Figure 16 compares the fitness of inter-story responses between three meta-models. It can be observed that the accuracy of both inter-story velocity and displacement responses slightly decreases compared with those of absolute responses. The Kriging-NARX model shows a significant reduction in the accuracy of upper floors compared with PCE/APC-NARX models due to the error accumulation at the end of time history analysis. Good fitness of inter-story velocity can still be observed between Kriging/PCE/APC-NARX model prediction and MCS results.

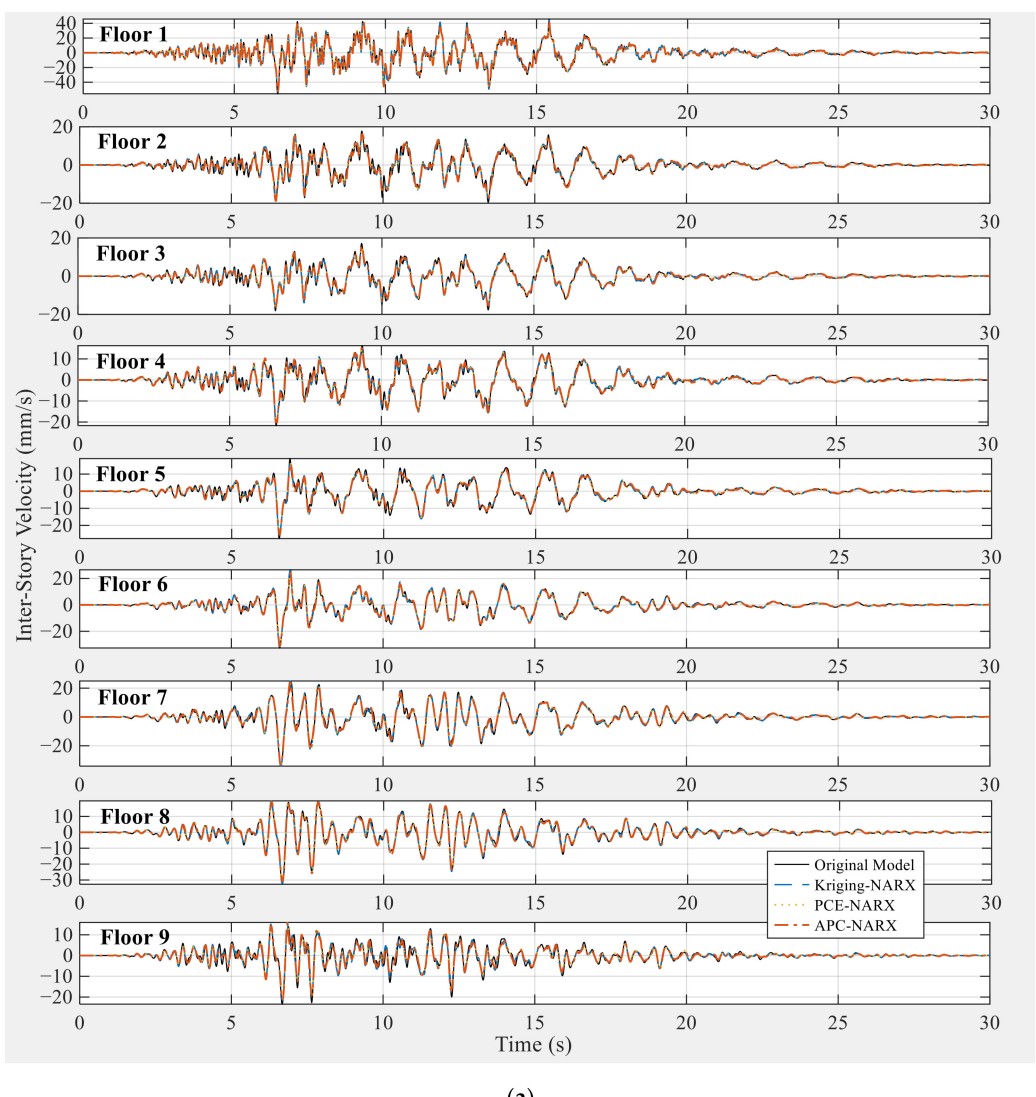

(**a**)

**Figure 15.** *Cont*.

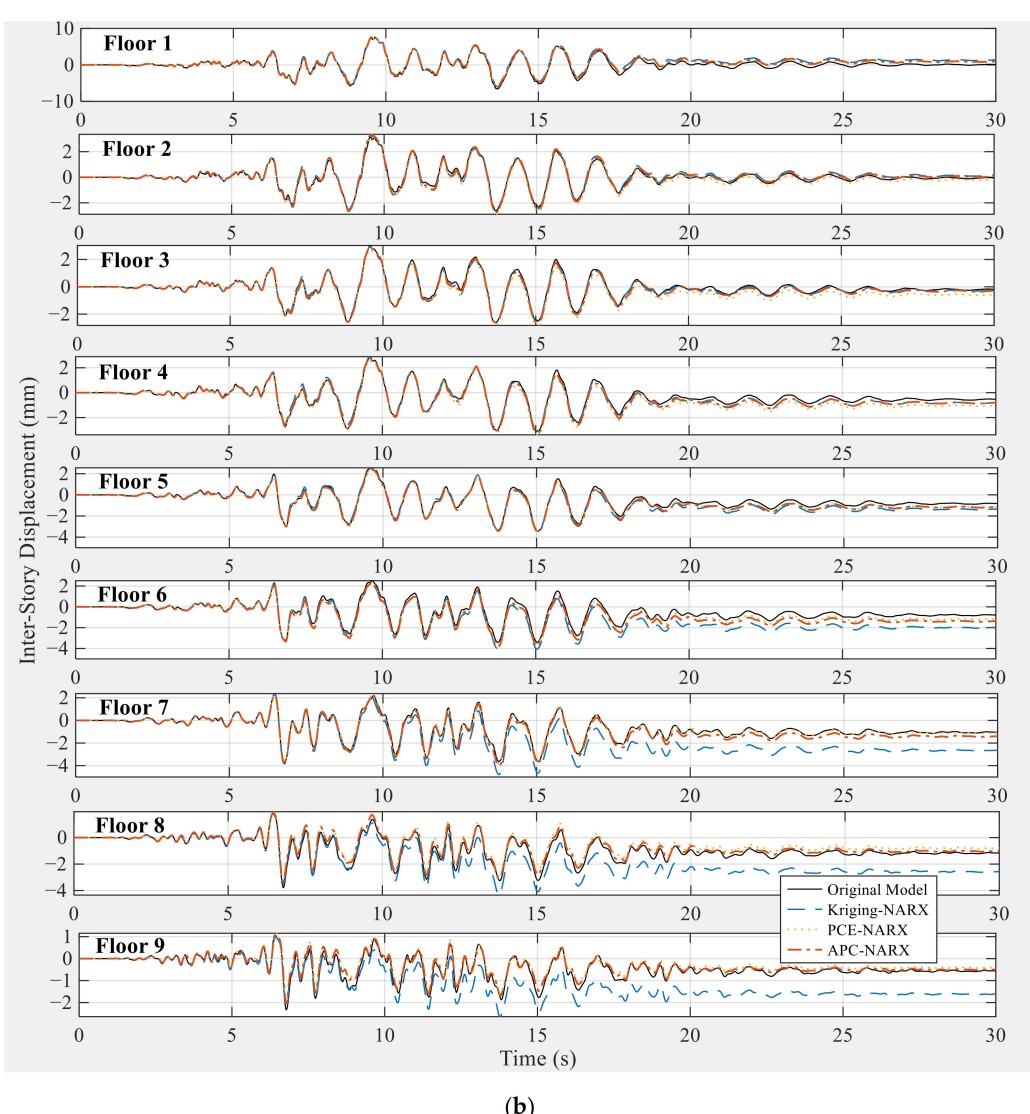

**(b)**

**Figure 15.** Time history of the inter-story responses (a typical sample). (**a**) Velocity responses; (**b**) Displacement responses.

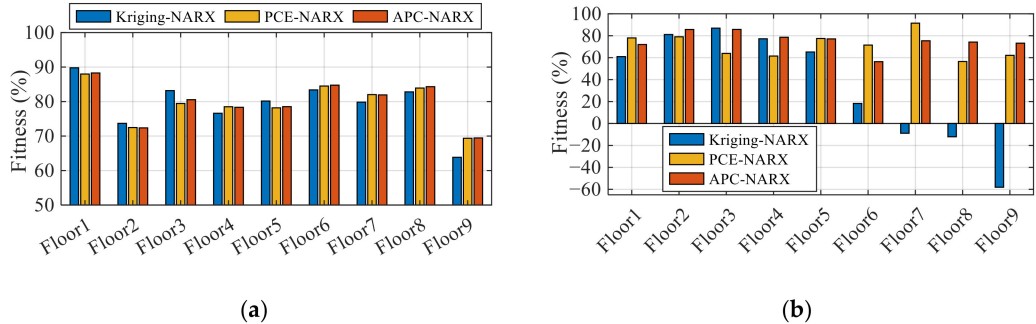

**Figure 16.** Fitness for the inter-story responses (a typical sample). (**a**) Velocity responses; (**b**) Displacement responses.

### 5.4. Comparison of Peak Story Responses

The safety and serviceability requirements for structural design mainly depend on the maximum peak responses. Figure 17a compares the peak inter-story velocity responses between Kriging/PCE/APC-NARX model prediction and MCS. It can be found that peak values from Kriging/PCE/APC-NARX models align well with actual values from MCS

validation cases. The peak values of inter-story displacement responses are further compared in Figure 17b, where good agreement again can be observed. It demonstrates that the maximum peak displacements are generally close to each other and are not significantly affected by the accumulation errors from integrating the time history of velocity responses. Correlation coefficients between the actual and predicted maximum peak values of velocity and displacement responses are further compared in Figure 18. It should be noticed that the correlation coefficients are relatively low due to some extreme data points commonly referred to as outliers. Figure 18 implies that the correlation coefficients for peak displacement are lower than those of peak velocity due to the existence of more outliers. For peak responses, it can be further observed that PCE and APC meta-models have more stable performance than the Kriging meta-models.

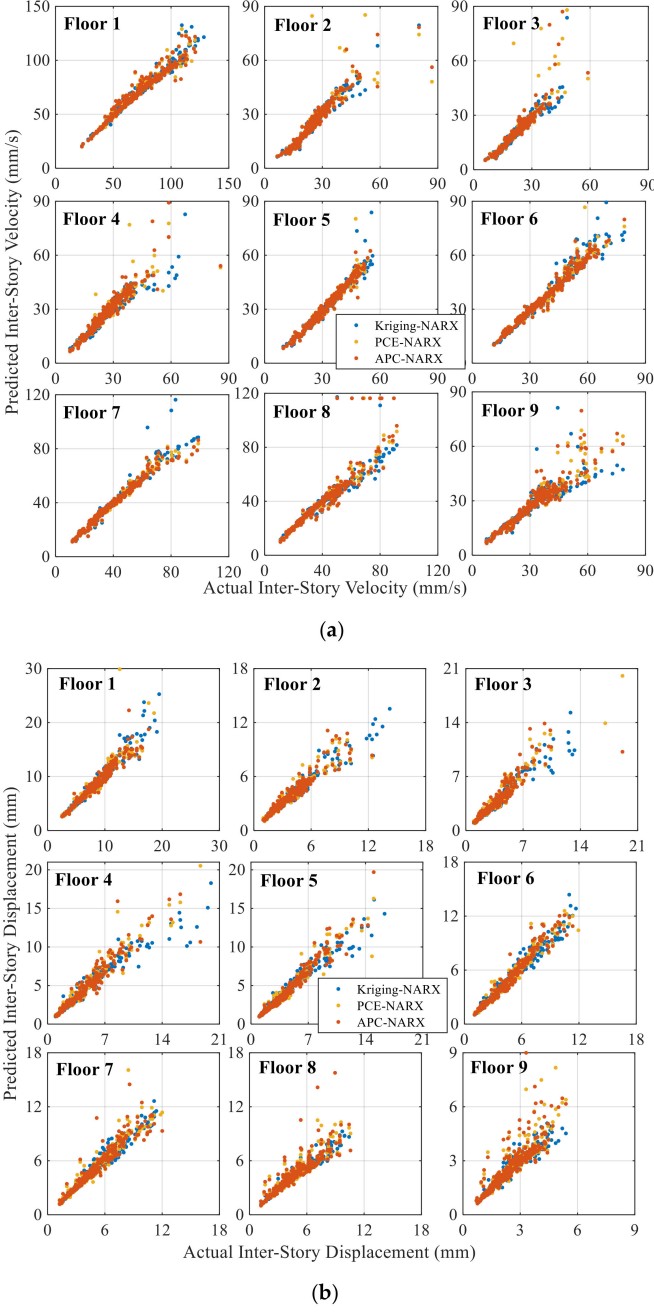

**Figure 17.** Peak values of velocity responses (300 samples). (**a**) Peak velocity; (**b**) Peak displacement.

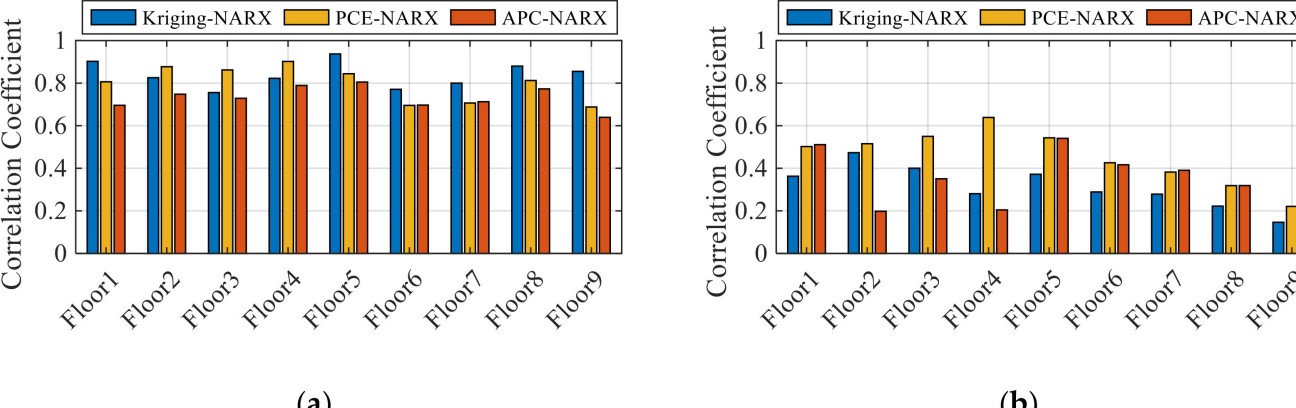

**Figure 18.** Correlation coefficients for peak responses (300 samples). (**a**) Peak velocity; (**b**) Peak displacement.

### 5.5. Statistics for All Validation Cases

The statistical information of the total 300 validation samples is further summarized for comparison of the Kriging/PCE/APC-NARX models and traditional MCS in Figure 19. Figure 19a shows the box plot for the mean fitness of the four outputs in reduced coordinates, in which the red line indicates the median, the blue box indicates the 25th and 75th percentiles, and the red plus symbol indicates outliers. It demonstrates that all three meta-model-surrogated MIMO-NARX models have nearly similar accuracy for most MCS validation cases except for some outliers. Figure 19b shows the histogram with a distribution fitting curve of the mean fitness of the four outputs in reduced coordinates, which illustrates that APC-NARX has more counts for higher fitness than the corresponding Kriging- and PCE-NARX models. Figure 19c–f shows the statistical information for the mean fitness of the nine-DOF absolute velocity and displacement responses. It can be observed from Figure 19c,e that the Kriging/PCE/APC-NARX strategy is quite accurate in velocity prediction while the accuracy of displacement obtained by integration decreases to a certain extent. Figure 19d,f indicates that PCE- and APC-NARX are superior to Kriging-NARX in terms of the counts of good fitness. Figure 19g–j shows the statistical information for the mean fitness of the nine inter-story responses of velocity and displacement. The accuracy is observed to decrease when compared with those in Figure 19c–f. Lower accuracy brings more outliers, which interferes with the comparison of the three meta-model surrogated MIMO-NARX models.

Figure 20 shows the variation in the accuracy of Kriging/PCE/APC-NARX model prediction with respect to the mean ductility ratio of the nine inter-story Bouc–Wen hysteresis. The fitness and ductility in Figure 20 represent the mean value of the fitness and ductility ratio of the four outputs or nine responses. As shown in Figure 20a–e, the accuracy of the Kriging/PCE/APC-NARX strategy significantly decrease with the increase of inter-story ductility. Figure 20c,e illustrates this phenomenon again that the predicted accuracy of displacement is generally inferior to that of velocity due to the accumulation error. It can be demonstrated that the Kriging/PCE/APC-NARX strategy is basically accurate for most MCS validation cases within a certain nonlinearity range.

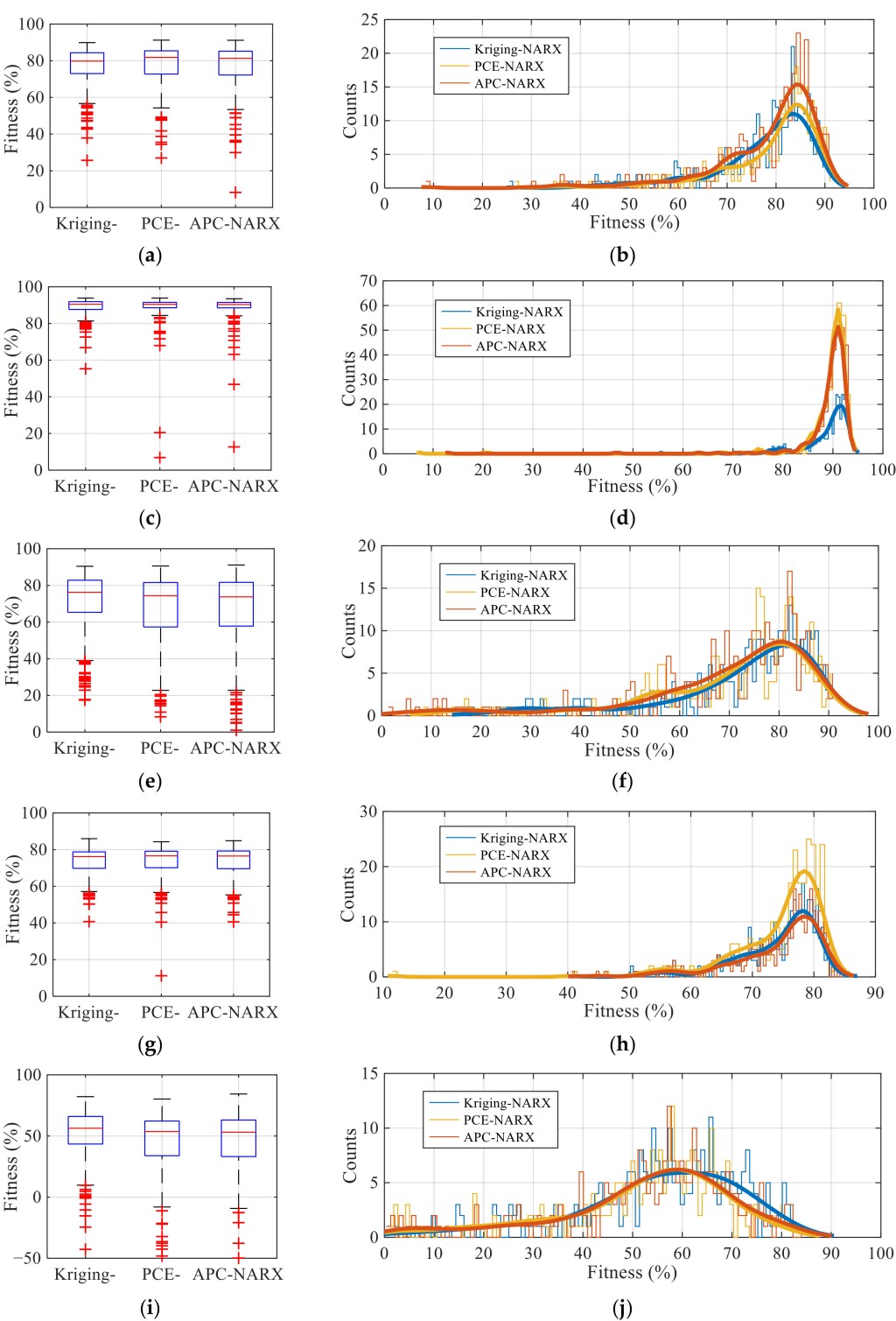

**Figure 19.** Boxplot and histogram of the mean fitness of different time history curves (300 samples). (**a**) The mean fitness of the four outputs in reduced coordinates: Boxplot; (**b**) Histogram; (**c**) The mean fitness of the nine-DOF absolute velocity responses: Boxplot; (**d**) Histogram; (**e**) The mean fitness of the nine-DOF absolute displacement responses: Boxplot; (**f**) Histogram; (**g**) The mean fitness of the nine inter-story velocity responses: Boxplot; (**h**) Histogram; (**i**) The mean fitness of the nine inter-story displacement responses: Boxplot; (**j**) Histogram.

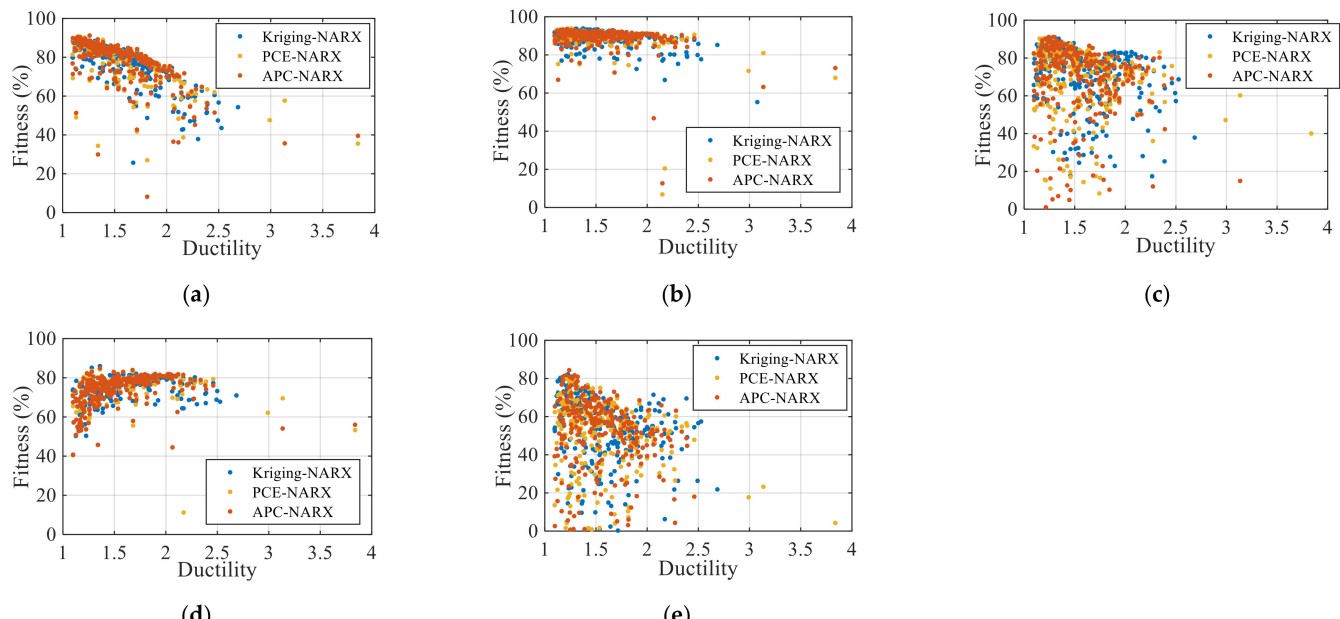

**Figure 20.** Relationship between the mean ductility ratio and the mean fitness (300 samples). (**a**) Outputs in reduced coordinates; (**b**) Absolute velocity responses; (**c**) Absolute displacement responses; (**d**) Inter-story velocity responses; (**e**) Inter-story displacement responses.

## 6. Conclusions

This research presents a computational study of different surrogate strategies to account for uncertainty propagation in nonlinear MDOF structural systems. A nine-DOF structure is selected for the numerical study for generating the training and validation samples. MIMO-NARX modeling is established to capture the nonlinear dynamic behavior in MOR-reduced coordinates using the results from dynamic time history simulations. The NARX coefficients are surrogated using meta-models including Kriging, PCE, and APC with uncertain variables of ground motion and structural mass.

The MCS results show good fitness of responses between MIMO-NARX models and the traditional dynamic simulation using the integration algorithm. It is worth noting that the absolute velocity responses are used in NARX training for improving the nonlinear capture effectiveness. The accuracy of the four reduced coordinates decreases with the increase of frequency due to difficulty in capturing high-frequency nonlinearity. Good performances are observed for all three meta-models in absolute velocity response prediction when compared with MCS. Generally, a slight decrease in accuracy is observed in displacement response prediction obtained by integrating the velocity responses due to the drift errors caused by error accumulation at the end of time history analysis. The accuracy of both inter-story velocity and displacement responses show different degrees of decrease compared with absolute velocity and displacement responses. The integral and accumulation drift errors do not necessarily affect the meta-model predicted peak displacement responses, which are more concerned with structural safety and serviceability. From a statistical point of view, all three meta-models surrogated MIMO-NARX models have nearly similar accuracy for most of the three hundred MCS validation cases except for some outliers. PCE- and APC- are shown to be superior to Kriging-NARX in terms of the counts of good fitness according to some statistical information. Considering the advantages of APC over PCE in the scope of application, APC-NARX is best suitable to replace the original model in a large number of MCS. It is illustrated that the MIMO-NARX modeling with surrogated coefficients by meta-models provides an effective and efficient tool for uncertainty quantification of nonlinear structures in earthquake engineering.

This study takes a nine-DOF structure as a proof of concept to verify the effectiveness of the proposed method. The nonlinear performance of the Bouc–Wen model in the MDOF

system changes with different stochastic seismic excitations and structural properties. The choice of the basic functions in MIMO-NARX, which depends on different nonlinear types of structural models, remains to be studied in the future.

It should be noticed that the accuracy of the MIMO-NARX model is affected and limited by the intensity of nonlinearity. Although the accuracy of the three meta-model-surrogated MIMO-NARX models is significantly decreased with the intensity of structural ductility increase, it still can be demonstrated that the Kriging/PCE/APC-NARX strategy is basically accurate for most of the MCS validation cases within a certain nonlinearity range. Future research may focus on finding more suitable NARX basis functions to eliminate the adverse effect of high nonlinearity on accuracy.

**Author Contributions:** Conceptualization, C.C.; methodology, C.C., X.G. and M.C.; software, X.G. and M.C.; validation, M.C.; formal analysis, M.C.; investigation, X.G. and M.C.; resources, X.G. and M.C.; data curation, M.C.; writing—original draft preparation, M.C.; writing—review and editing, C.C.; visualization, M.C.; supervision, T.G.; project administration, T.G.; funding acquisition, T.G. and W.X. All authors have read and agreed to the published version of the manuscript.

**Funding:** This research was funded by the Ministry of Science and Technology of the People's Republic of China (Grant No. 2018YFE0206100), the National Natural Science Foundation of China (Grant No. 52178114 and No. 51808111), Young scientific and technological talents promotion project of Jiangsu Association for science and technology (No. 2021-79).

**Institutional Review Board Statement:** Not applicable.

**Informed Consent Statement:** Not applicable.

**Data Availability Statement:** Not applicable.

**Conflicts of Interest:** The authors declare no conflict of interest.

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
