# Peer review of "A Comparative Study of Meta-Modeling for Response Estimation of Stochastic Nonlinear MDOF Systems Using MIMO-NARX Models"

_applsci, doi:10.3390/app122211553_

Round 1

Reviewer 1 Report

This paper investigated the meta-modeling for response estimation of stochastic nonlinear MDOF systems using MIMO-NARX Models. The English writing of the paper is acceptable, however, it is recommended to revise it for minor grammatical issues including prepositions, punctuation, etc. The introduction section is written well including a good literature review and statement of the problem. The theory of the porous medium and the proposed method was explained well, enough formulation was provided as well as verification examples. Hence, it can be considered for publication.

Reviewer 2 Report

This manuscript proposed MIMO-NARX models alongside a model order reduction and different meta-modeling techniques for dynamic behavior of nonlinear structures and uncertainty analysis. A simple numerical model was used to validate the proposed method. Generally, the manuscript is very well-written with sufficient details, discussions, and results. I am positive for publication after some revisions.

1)        The major innovations and originality of this research were not explained properly.

2)        Please further discuss the limitations of the method used in this research.

3)        Validation of the method considered in this research by only one simple numerical model is subjective! The authors may add another complex structure, preferably an experimental model if available or a 2D or 3D asymmetric numerical structure, for improving and demonstrating the effectiveness of the method.

4)        Please manage the number of figures. Some of them can be merged into a single figure.

5)        Some corrections should be carried out to improve English grammar errors such as better uses of English articles a/an/the (i.e., in a few sentences) and single/plural nouns such as Line 63 (A NARX model), Lines 482, 486, 487, 489 (i.e., please use Figures # and #), etc.

Reviewer 3 Report

- Application of MIMO-NARX models into uncertainty analysis in structure engineering.

- Nowadays, there are no 100% original themes.

- Application of MIMO-NARX models into uncertainty analysis in structure engineering.

- The authors should perform experimental verification of their proposed method.

- Conclusions are consistent with the evidence and arguments presented and they address the main posed question.

- The references are appropriate.

- Figures and tables are clear and readable.
